# Comorbid Asthma Increased the Risk for COVID-19 Mortality in Asia: A Meta-Analysis

**DOI:** 10.3390/vaccines11010089

**Published:** 2022-12-30

**Authors:** Liqin Shi, Jiahao Ren, Yujia Wang, Huifen Feng, Fang Liu, Haiyan Yang

**Affiliations:** 1Department of Epidemiology, School of Public Health, Zhengzhou University, Zhengzhou 450001, China; 2School of Nursing, Hebi Polytechnic, Hebi 458030, China; 3Department of Infectious Diseases, The Fifth Affiliated Hospital of Zhengzhou University, Zhengzhou 450052, China

**Keywords:** asthma, coronavirus disease 2019, mortality, meta-analysis, Asia

## Abstract

We aimed to explore the influence of comorbid asthma on the risk for mortality among patients with coronavirus disease 2019 (COVID-19) in Asia by using a meta-analysis. Electronic databases were systematically searched for eligible studies. The pooled odds ratio (OR) with 95% confidence interval (CI) was estimated by using a random-effect model. An inconsistency index (I^2^) was utilized to assess the statistical heterogeneity. A total of 103 eligible studies with 198,078 COVID-19 patients were enrolled in the meta-analysis; our results demonstrated that comorbid asthma was significantly related to an increased risk for COVID-19 mortality in Asia (pooled OR = 1.42, 95% CI: 1.20–1.68; I^2^ = 70%, *p* < 0.01). Subgroup analyses by the proportion of males, setting, and sample sizes generated consistent findings. Meta-regression indicated that male proportion might be the possible sources of heterogeneity. A sensitivity analysis exhibited the reliability and stability of the overall results. Both Begg’s analysis (*p* = 0.835) and Egger’s analysis (*p* = 0.847) revealed that publication bias might not exist. In conclusion, COVID-19 patients with comorbid asthma might bear a higher risk for mortality in Asia, at least among non-elderly individuals.

## 1. Introduction

Coronavirus disease 2019 (COVID-19), which is brought on by severe acute respiratory syndrome coronavirus 2 (SARS-CoV-2), has ravaged the world. As of 26 October 2022, 624 million patients have been confirmed with the COVID-19 diagnosis globally of which 6.5 million patients have died [1]. Vaccines have shown to be very effective against severe COVID-19 disease and mortality [2,3,4,5,6,7]; it is also important to understand risk factors (e.g., to decide whom to prioritize for vaccination). Until now, several variables (age, sex, and certain past medical history) have been identified as risk factors for COVID-19 mortality [8,9,10,11,12,13]. Although there have been several meta-analyses exploring the relationship of comorbid asthma with the risk for COVID-19 mortality in the full regions [14,15,16,17,18], the findings from previous meta-analyses were still inconclusive, which might suffer limitations from considerable variability in the prevalence of asthma across different regions [14,19,20]. Therefore, it is an urgent requirement to investigate the relationship of comorbid asthma with the risk for COVID-19 mortality based on specific regions.

To the best of our knowledge, three meta-analyses have explored this relationship in Asia [15,18,21]. However, the number of included studies (all are less than twelve) and the sample sizes are limited. Moreover, the conclusions drawn from these articles are inconsistent or even contradictory. Additionally, a substantial number of articles on this topic in Asia have emerged since then. Taken together, we conducted this updated meta-analysis to ascertain the relationship between comorbid asthma and COVID-19 mortality in Asia on the basis of the latest data.

## 2. Methods

### 2.1. Search Strategy and Literature Management

This quantitative meta-analysis was performed according to the statement of PRISMA (preferred reporting items for systematic reviews and meta-analyses). A systematic literature search was undertaken among electronic databases containing PubMed, Scopus, EMBASE, Springer, Web of Science, and Wiley to recognize eligible studies from inception to 22 October 2022. Searching strategies were as follows: (“COVID-19” OR “coronavirus disease 2019” OR “SARS-CoV-2” OR “2019-nCoV” OR “novel coronavirus”) and (“asthma” OR “bronchial asthma”) and (“mortality” OR “non-survivor” OR “fatality” OR “deceased” OR “death”). Additionally, to achieve extensive searches, relevant references of included studies and reviews were also taken into consideration.

### 2.2. Selection Criteria

Studies were selected if they were amenable to the following criteria: (1) Adult COVID-19 patients should be diagnosed in line with the World Health Organization (WHO) guidance. (2) Studies were conducted in Asia and explicitly reported the number of COVID-19 patients with comorbid asthma and outcome of interest (alive or dead) or the effect size with 95% confidence interval (CI) concerning the relationship of asthma with COVID-19 mortality. (3) Articles should be written in English. Studies based on criteria as follows must be cast off accordingly: (1) preprints, comments, errata, reviews, and repeated articles. (2) articles without available data concerning the incidence of asthma and death among patients with COVID-19 in Asia.

### 2.3. Data Extraction

Two researchers respectively inspected all the literature depending on the criteria of inclusion and exclusion and then extracted the relevant information, including author, male proportion, country, cases, study design, setting, mean age with standard deviation or median age with interquartile range, incidence of non-survivors and survivors among patients with COVID-19 and comorbid asthma and those without, or the effect size with corresponding 95% CI. If two or more publications are sourced with the same author or the same institute, we then reviewed the time period of participant enrollment among the studies. If the time period of participant enrollment was the same or the study start and end times were crossed among the studies, we regarded these studies as having the same participants or overlapping participants; otherwise, we regarded these studies as different. For these studies based on the same data source, we included only the articles with the most complete data. If there was any disagreement, it was settled through a third investigator or by discussion to reach a consensus.

### 2.4. Statistical Analysis

All the statistical analyses were implemented on STATA (Version 16) and R software (Version 4.2.1) with attached “meta” package (Version 5.5-0). Pooled OR and 95% CI were computed by a random-effect model to describe the relationship of asthma with COVID-19 mortality in Asia. Two tailed *p*-value less than 0.05 was considered as statistical significance. An inconsistency index (I^2^) was applied to evaluate the statistical heterogeneity among studies [22]. Meta-regression and subgroup analyses were undertaken to find possible sources of heterogeneity. To test the stability of our study, a sensitivity analysis by omitting one single study at a time was carried out. Both Begg’s analysis and Egger’s analysis were implemented to test the potential publication bias [23,24].

## 3. Results

### 3.1. Study Selection

Online literature searches yielded 22,361 citations from electronic databases, and an additional 50 records were found from the references of cited lists. After removing 19,168 duplicates, 3243 articles were initially identified. Next, 2943 articles were excluded after reading the abstracts. After that, 300 articles were evaluated for full-text eligibility, and 197 articles with available data but not Asian were excluded. Ultimately, 103 studies conducted in Asia were enrolled in this meta-analysis [25,26,27,28,29,30,31,32,33,34,35,36,37,38,39,40,41,42,43,44,45,46,47,48,49,50,51,52,53,54,55,56,57,58,59,60,61,62,63,64,65,66,67,68,69,70,71,72,73,74,75,76,77,78,79,80,81,82,83,84,85,86,87,88,89,90,91,92,93,94,95,96,97,98,99,100,101,102,103,104,105,106,107,108,109,110,111,112,113,114,115,116,117,118,119,120,121,122,123,124,125,126,127]. The flow chart of the literature search and selection process is illustrated in Figure 1.

### 3.2. Descriptive Characteristics

Summary characteristics of the enrolled studies are tabulated in Table 1. This meta-analysis was based on a total of 103 eligible studies with 198,078 COVID-19 patients. In terms of study design, there were eighty-five retrospective studies, ten prospective studies, six cross-sectional studies, one case series study, and one clinical trial study. From the point of view of geographical settings, we characterized the country’s levels of social and economic development based on the 4-tier Human Development Index (HDI) from the United Nation’s 2022 Human Development Report. Additionally, countries were categorized as very high HDI country (Korea, Singapore, Israel, Japan, Turkey, Kuwait, Saudi Arabia, and United Arab Emirates), high HDI country (China, Iran, and Indonesia), medium HDI country (India, Philippines, and Bangladesh), and low HDI country (Pakistan). Among these studies, ninety-eight studies reported the exact numbers of non-survivors and survivors of COVID-19 patients with asthma, while five studies reported OR with 95% CI to reflect the effect of comorbid asthma on Asian COVID-19 mortality.

### 3.3. Asthma and COVID-19 Mortality in Asia

Overall, combining the data from 103 studies, our meta-analysis indicated there was a significant association between comorbid asthma and increased risk for mortality of COVID-19 patients (pooled OR = 1.42, 95% CI: 1.20–1.68; I^2^ = 70%, *p* < 0.01, Figure 2). Consistent results were observed in the subgroup analyses stratified by sample sizes (pooled OR = 1.41, 95% CI: 1.10–1.82 for <1000 cases and 1.43, 95% CI: 1.14–1.79 for ≥1000 cases), setting (pooled OR = 1.37, 95% CI: 1.14–1.64 for hospitalized patients and 1.91, 95% CI: 1.36–2.68 for all patients) and male proportion (pooled OR = 2.08, 95% CI: 1.78–2.44 for <50% and 1.24, 95% CI: 1.00–1.55 for ≥50%). When the subgroup analysis was performed by age, the significant relationship existed in the subgroup of mean/median age <60 years old (pooled OR = 1.44, 95% CI: 1.18–1.76) but did not exist in the subgroup of mean/median age ≥60 years old (pooled OR = 1.36, 95% CI: 0.95–1.94). The significant association existed among studies with the prevalence of obesity ≥20% (pooled OR = 2.27, 95% CI: 1.65–3.12) but did not exist among studies with the prevalence of obesity <20% (pooled OR = 1.21, 95% CI: 0.60–2.46). The subgroup analysis according to ICU and non-ICU patients suggested COVID-19 patients with asthma had a significantly increased risk of mortality among studies with non-ICU patients (pooled OR = 1.45, 95% CI: 1.22–1.72) but not among studies with ICU patients (pooled OR = 1.40, 95% CI: 0.58–3.38). A further subgroup analysis by country characterized by homogenous socioeconomic features revealed COVID-19 patients with asthma had a significantly increased risk for mortality compared with patients without asthma among very high HDI countries (pooled OR = 1.55, 95% CI: 1.29–1.87) but not among high HDI countries (pooled OR = 1.38, 95% CI: 0.91–2.11), medium HDI countries (pooled OR = 0.88, 95% CI: 0.50–1.56), and low HDI countries (pooled OR = 1.60, 95% CI: 0.77–3.33) (as shown in Figure 2). The meta-regression displayed male proportion (*p* = 0.034) might be the potential sources of heterogeneity, while country (*p* = 0.301), sample sizes (*p* = 0.966), setting (*p* = 0.254), age (*p* = 0.961), and obesity (*p* = 0.103) might not.

Considering comorbidities could lead to additional ”noise” and measurement error, we subsequently calculated the pooled OR on the basis of adjusted effect estimates. The results indicated asthma was significantly associated with the increased risk of mortality among Asian COVID-19 patients on the basis of 19 studies (adjusted OR = 1.22, 95% CI: 1.05–1.42), which supported the findings based on crude effects.

### 3.4. Sensitivity Analysis and Publication Bias

The sensitivity analysis showed the effect estimate was not unduly impacted by any single study, indicating the stability and robustness of our results. Begg’s analysis (*p* = 0.835, Figure 3A) and Egger’s analysis (*p* = 0.847, Figure 3B) demonstrated publication bias might not exist in this study.

## 4. Discussion

Our findings based on 103 eligible studies indicated comorbid asthma was significantly associated with an increased risk for mortality in COVID-19 patients compared with those without in Asia. The subgroup analyses by male proportion, sample sizes, and setting yielded consistent results, but the subgroup analysis by age indicated comorbid asthma was significantly associated with higher risk for COVID-19 mortality in Asia among studies with mean/median age <60 years old and not among studies with mean/median age ≥60 years old. Previous studies have shown advanced age and asthmatic patients are prone to other comorbidities, such as hypertension and diabetes mellitus, which are closely related to the severity and mortality of COVID-19 patients [128]. We also investigated the proportion of hypertension and diabetes mellitus in enrolled studies among age <60 years old and ≥60 years old and found the proportions of hypertension and diabetes mellitus were relatively higher in groups ≥60 (44.39% and 30.64%, respectively) than in groups <60 years old (32.03% and 25.8%, respectively). Thus, we speculated the existence of other comorbidities (such as hypertension and diabetes mellitus) might mask the relationship between asthma and COVID-19 mortality.

Asthma is a heterogeneous disease, and some phenotypes are related to obesity, which has continued to attract respiratory experts’ attention since the pandemic of COVID-19 [129]. Furthermore, high body mass index has been identified as a risk factor for COVID-19 mortality. In our subgroup analyses stratified according to the proportion of males, the odds ratio increased to 2.08 in the group with males being less than 50% and was reduced to 1.24 in the subgroup where males dominated, while Wenzel et al. showed females usually dominate in obesity-related asthma [129]. This suggested part of the conflicting results on asthma and COVID-19 mortality could be due to the differences in handling of obesity in different studies. Our further analysis stratified by obesity prevalence supported this opinion. The subgroup results regarding ICU versus non-ICU deaths may be explained by the facts that ICU patients may rely either on a better economic status (theirs or for their countries) than non-ICU patients (whose access to intensive care may be impaired by economic status).

At present, research has explored the potential mechanism of the association between COVID-19 and asthma from the standpoints of pathophysiology. Viral infections, including SARS-CoV-2 and Middle East respiratory syndrome coronavirus (MERS) could directly result in the exacerbation of asthma and thus lead to serious airway inflammation, which might be linked to critical unfavorable outcomes [130,131]. Additionally, several researchers proposed host antiviral immunity was decreased due to asthma associated type II inflammatory response [132,133]. In addition, interferon responses as a crucial step of antiviral immune reaction were shown to be lacking in asthmatic patients attributed to decreased production [134,135]. Additionally, asthma resulted in mucus plugging in the lower respiratory tract, limiting the airflow and worsening the hypoxemia from diffuse alveolar damage by SARS-CoV-2 infection [136]. However, these theoretical relationships remain to be observed. Further studies focusing on the molecular mechanisms underlying the association between comorbid asthma and the increased risk for COVID-19 mortality are warranted to verify our results.

The strengths of this study were the number of eligible studies included (103 eligible studies), and the sample sizes (198,078 cases) were large, and subgroup analyses were conducted. However, several limitations need to be acknowledged in this study. First, although the investigators tried to avoid duplicates in the process of article selection, several studies collected clinical records of COVID-19 patients retrospectively from large national public databases. This inevitably led to repeated populations and selective bias. Second, most of the included articles were observational retrospective studies, which resulted in lack of proof for casual links between asthma and mortality risk of COVID-19. Third, the medical history of enrolled patients could not be clearly analyzed, especially for the use of theophylline, inhaled corticosteroids, leukotriene receptor antagonists, and other drugs. Length of in-hospital treatment, the severity of asthma, types of asthma, and other comorbidities could not be extracted either, which should be a focus in the future study. Fourth, the existence or history of other comorbidities, such as coronary artery disease, chronic obstructive pulmonary disease, and so on was not addressed presently, which still restricted the generalization of our findings.

## 5. Conclusions

Our findings demonstrated comorbid asthma significantly increased the risk for mortality among patients with COVID-19 in Asia, at least among non-elderly individuals.

## Figures and Tables

**Figure 1 vaccines-11-00089-f001:**
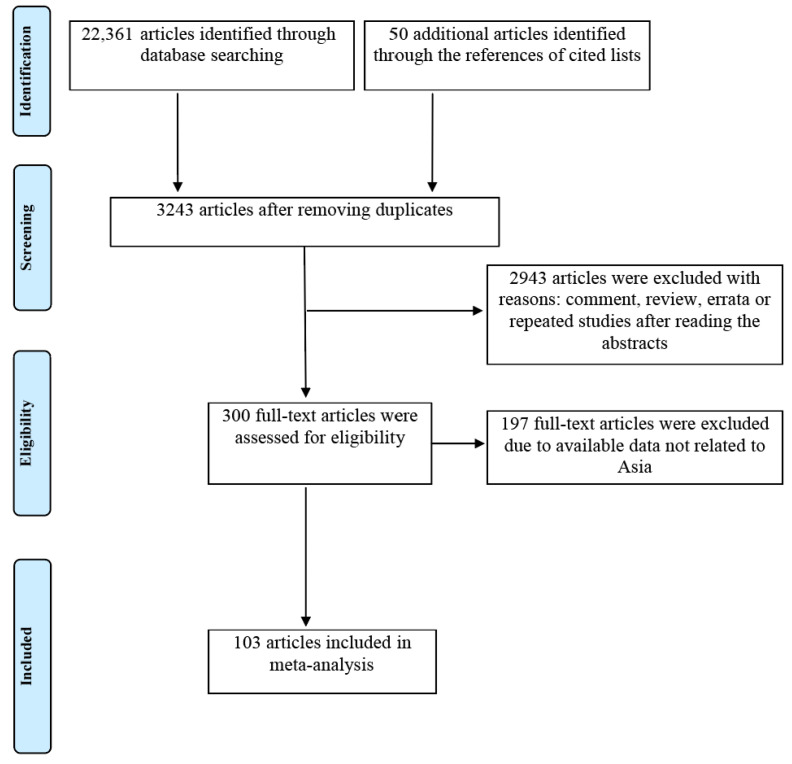
Flow chart of study search and selection process.

**Figure 2 vaccines-11-00089-f002:**
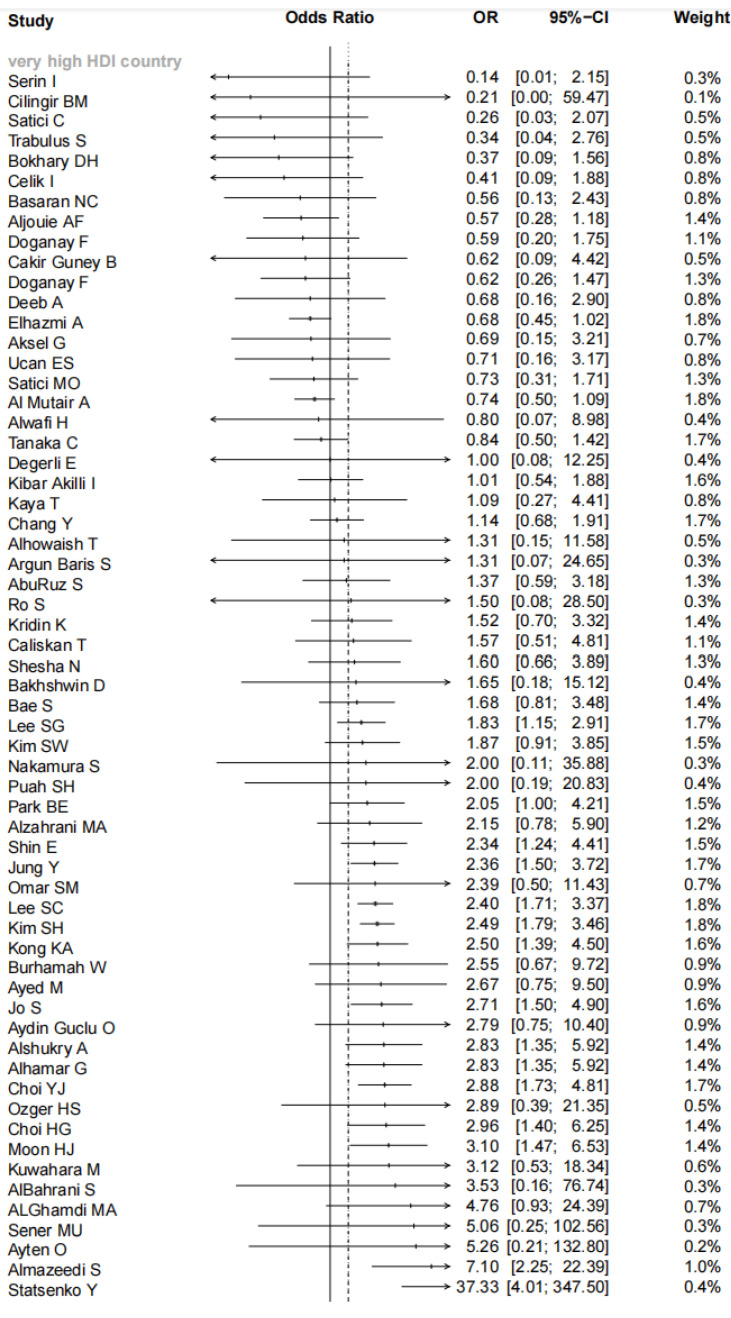
Forest plot indicated there was a significant association between comorbid asthma and the increased risk for mortality of COVID-19 patients in Asia.

**Figure 3 vaccines-11-00089-f003:**
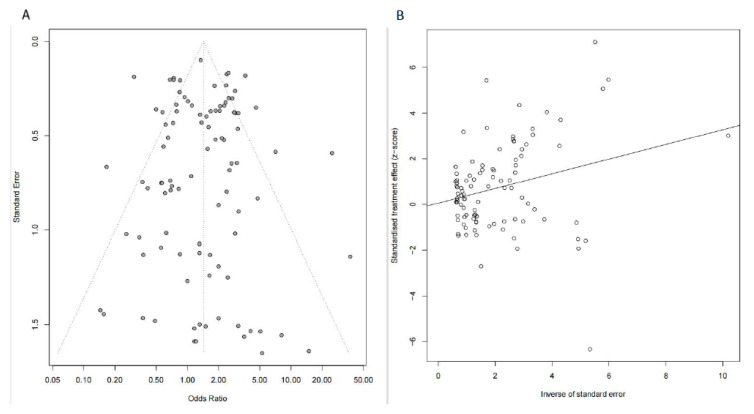
Publication bias was evaluated by Begg’s analysis (**A**) and Egger’s analysis (**B**).

**Table 1 vaccines-11-00089-t001:** Main characteristics of the included studies.

Author	Country	Study Design	Setting	Cases	Male (%)	Mean/Median Age	Asthma		Non-Asthma		Comorbidity	
							Dead	Alive	Dead	Alive	Hypertension	Diabetes
Lee SC [90]	Korea	Retrospective	All patients	7272	40.3	45.3	44	642	183	6403	19.3	14.3
Choi YJ [60]	Korea	Retrospective	All patients	7590	40.8	46.6 (27.1–61)	17	201	210	7162	NA	NA
Trabulus S [123]	Turkey	Retrospective	Hospitalized	336	57.1	55.0 ± 16.0	1	19	42	274	35.7	18.8
Aksel G [30]	Turkey	Prospective	Hospitalized	168	53.6	59.5 (48.3–76)	2	12	30	124	48.2	25.6
Serin I [117]	Turkey	Retrospective	All patients	2217	53.0	47.66 ± 17.23	0	103	68	2046	20.6	16.2
Ayaz A [46]	Pakistan	Retrospective	Hospitalized	66	61.0	50.6 ± 19.1	0	2	9	55	45.5	37.9
Ayed M [48]	Kuwait	Retrospective	ICU patients	103	85.5	53 (44–63)	8	4	39	52	35.0	39.2
Lee SG [91]	Korea	Retrospective	All patients	7339	40.1	47.1 ± 19.0	21	376	206	6736	18.6	11.6
Choi HG [59]	Korea	Retrospective	Hospitalized	4057	42.5	54.1	8	88	118	3843	20.4	12.1
Zhou S [127]	China	Retrospective	Hospitalized	134	63.4	59.04 ± 17.74	14	6	58	56	28.4	13.4
Omar SM [96]	Saudi Arabia	Retrospective	Hospitalized	88	81.8	62 (55–70)	4	3	29	52	25.0	20.5
Caliskan T [56]	Turkey	Retrospective	Hospitalized	565	NA	48 ± 19.66	4	17	71	473	22.7	12.7
Kim SW [83]	Korea	Retrospective	Hospitalized	2254	35.8	58 (42–70)	9	57	170	2018	28.7	16.6
Park BE [100]	Korea	Retrospective	Hospitalized	2269	35.9	55.5 ± 20.2	9	58	155	2047	28.8	17.0
Alwafi H [42]	Saudi Arabia	Retrospective	Hospitalized	706	68.5	48.0 ± 15.6	OR (95% CI): 0.80 (0.07–8.82)				30.2	36.0
Kridin K [87]	Israel	Retrospective	Hospitalized	3618	39.7	38.6 ± 17.7	8	504	32	3074	NA	NA
Kim SH [82]	Korea	Retrospective	All patients	7590	40.8	45.87 ± 19.77	48	716	179	6647	NA	13.9
Bae S [50]	Korea	Retrospective	Hospitalized	1760	63.6	60.9 ± 18.6	9	52	159	1540	33.1	19.3
Moon HJ [94]	Korea	Retrospective	Hospitalized	4426	42.1	51 (30.2–63.7)	8	92	118	4208	21.4	12.3
Kong KA [85]	Korea	Retrospective	Hospitalized	5307	40.8	52.1 (33.7–64.5)	13	113	228	4953	22.6	12.9
Akhtar H [29]	Pakistan	Retrospective	Hospitalized	659	68.6	53.8	53	11	416	179	57.2	50.2
Al Mutair A [31]	Saudi Arabia	Retrospective	ICU patients	1470	74.0	55.9 ± 15.1	45	83	569	773	46.0	52.4
Sehgal T [115]	India	Prospective	Hospitalized	68	63.2	48 (20–85)	0	2	9	57	22.1	20.6
Rai D [107]	India	Retrospective	Hospitalized	984	77.4	50.73 ± 16.50	16	21	238	709	31.1	33.5
Jung Y [77]	Korea	Retrospective	Hospitalized	4066	37.5	53.38	24	338	108	3596	29.2	NA
Kolivand P [84]	Iran	Prospective	Hospitalized	960	100.0	56.99 ± 6.71	7	16	117	820	NA	NA
Rehman S [108]	Pakistan	Retrospective	Hospitalized	2048	59.4	56 (18–88)	77	58	513	1400	47.6	29.7
Tanaka C [122]	Japan	Retrospective	Hospitalized	1529	79.1	66.69 ± 12.38	19	63	382	1065	48.7	35.8
Cakir Guney B [55]	Turkey	Retrospective	ICU patients	134	60.4	68.90 ± 15.67	2	2	80	50	56.0	33.6
Ong AN [97]	Philippines	Retrospective	Hospitalized	355	55.8	62.69 ± 12.21	5	22	85	243	74.6	NA
Kwok WC [89]	China	Retrospective	Hospitalized	4498	48.8	47	10	155	60	4273	21.0	11.4
Abrishami A [26]	Iran	Retrospective	Hospitalized	80	65.0	54.29 ± 15.21	1	6	12	61	25.0	15.0
Cilingir BM [61]	Turkey	Prospective	Hospitalized	162	62.3	56.98 ± 17.79	OR (95% CI): 0.214 (0.001–77.242)				NA	NA
AbuRuz S [27]	United Arab Emirates	Retrospective	Hospitalized	3296	76.3	44.3 ± 13.4	6	159	84	3047	28.6	27.4
Aydin Guclu O [47]	Turkey	Retrospective	Hospitalized	202	50.5	50.17 ± 19.68	OR (95% CI): 2.793 (0.750–10.402)				30.2	16.3
Cortez KJC [62]	Philippines	Retrospective	Hospitalized	280	36.1	48.4 ± 18.5	1	16	12	251	44.3	17.0
Kouhpeikar H [86]	Iran	Retrospective	Hospitalized	583	52.3	61.4 ± 0.9	12	4	61	506	26.6	13.9
He C [70]	China	Retrospective	Hospitalized	702	52.3	66.0 (58–73)	3	34	19	646	NA	25.2
Pramudita A [104]	Indonesia	Retrospective	Hospitalized	243	53.1	48.04 ± 14.43	0	6	32	205	32.5	20.6
Hesni E [71]	Iran	Retrospective	Hospitalized	27,256	53.7	53.34 ± 22.74	26	284	2620	24,326	12.7	7.4
Chang Y [58]	Korea	Retrospective	All patients	3122	30.7	NA	OR (95% CI): 1.14 (0.68–1.90)				32.8	14.8
Alam MT [33]	Pakistan	Retrospective	All patients	209	71.3	56 (50–65)	2	8	58	141	50.2	40.2
Araban M [44]	Iran	Retrospective	All patients	3181	47.2	52.6 ± 20.8	10	84	300	2787	14.8	16.2
Patgiri P [102]	India	Cross-sectional	Hospitalized	165	75.8	68.4 ± 6.9	1	2	38	124	37.6	24.2
Alimohamadi Y [37]	Iran	Retrospective	Hospitalized	3759	57.1	57.48 ± 17.27	8	111	305	3335	29.5	24.7
Shin E [120]	Korea	Retrospective	Hospitalized	5625	41.2	NA	11	108	230	5276	21.4	12.3
Basaran NC [52]	Turkey	Prospective	Hospitalized	368	46.5	57	2	29	37	300	38.0	24.2
Kibar Akilli I [81]	Turkey	Retrospective	Hospitalized	1511	58.2	60.1 ± 14.7	12	123	121	1255	48.0	33.3
Malundo AFG [93]	Philippines	Retrospective	Hospitalized	1215	52.5	55 (42–66)	9	78	212	916	48.0	25.6
Alhowaish T [36]	Saudi Arabia	Retrospective	Hospitalized	122	18.9	48.3 ± 16	1	6	13	102	32.0	27.9
Rohani-Rasaf M [110]	Iran	Cross-sectional	Hospitalized	1228	49.8	58.8 ± 16.2	8	80	80	1060	NA	23.7
Dana N [63]	Iran	Cross-sectional	Hospitalized	831	54.3	63.9 ± 16.2	OR (95% CI): 0.67 (0.08–5.41)				39.1	32.6
Jalili M [73]	Iran	Retrospective	Hospitalized	28,981	56.0	57.33 ± 17.67	141	432	5552	22,856	NA	11.3
Nakamura S [95]	Japan	Retrospective	Hospitalized	32	69.0	74.5 (24–90)	1	1	10	20	40.6	21.9
Saha A [112]	Bangladesh	Retrospective	ICU patients	168	79.8	56.26 (45.68–75.33)	3	12	92	61	41.1	52.4
Almazeedi S [40]	Kuwait	Retrospective	All patients	1096	81.0	41 (25–75)	4	39	15	1038	16.1	14.1
Alshukry A [41]	Kuwait	Retrospective	Hospitalized	417	63.0	45.39 ± 17.06	12	29	48	328	29.5	23.3
Jin M [75]	China	Retrospective	Hospitalized	121	33.9	57.52 ± 14.71	1	20	2	98	26.5	13.2
Rahimzadeh P [106]	Iran	Case series	ICU patients	70	66.0	66.22 ± 14.36	5	0	51	14	50.0	42.0
Zhang JJ [126]	China	Retrospective	Hospitalized	289	53.4	56 ± 11.56	1	0	48	240	28.0	9.3
Aljouie AF [39]	Saudi Arabia	Retrospective	Hospitalized	1513	56.8	54.83 ± 17.00	8	135	128	1242	40.0	40.2
Agrupis KA [28]	Philippines	Retrospective	Hospitalized	367	57.0	51 ± 18	0	15	60	292	38.1	20.2
Islam MA [72]	Bangladesh	Clinical trial	Hospitalized	199	79.0	64.0 (53.0–70.0)	2	0	75	122	77.9	9.5
Khalid A [79]	Pakistan	Retrospective	Hospitalized	317	62.5	NA	2	11	55	249	39.1	35.3
Safari M [111]	Iran	Retrospective	Hospitalized	66	60.6	61.6 ± 13.5	16	20	9	21	24.4	21.2
Pakdel F [99]	Iran	Cross-sectional	Hospitalized	15	66.0	47.25 ± 16.39	1	1	6	7	46.0	86.0
Satici C [113]	Turkey	Retrospective	Hospitalized	681	51.0	56.9 ± 15.7	1	42	54	584	34.4	28.0
Doganay F [66]	Turkey	Retrospective	Hospitalized	481	53.0	67 (52–79)	4	20	116	341	32.0	25.2
Ucan ES [124]	Turkey	Retrospective	Hospitalized	298	49.7	61.85 ± 20.01	2	16	42	238	45.6	16.8
Statsenko Y [121]	United Arab Emirates	Retrospective	ICU patients	72	80.6	58.66 ± 13.02	6	1	9	56	31.9	37.5
Burhamah W [54]	Kuwait	Retrospective	ICU patients	133	68.0	59 (49–68)	10	3	68	52	55.0	57.0
Khani M [80]	Iran	Prospective	Hospitalized	207	57.5	54.5 ± 14.8	0	10	22	175	38.2	25.1
Doganay F [67]	Turkey	Retrospective	Hospitalized	489	51.7	59.33 ± 19.42	7	24	147	311	36.6	26.0
Degerli E [65]	Turkey	Retrospective	Hospitalized	45	51.0	60.3 ± 15.65	2	1	28	14	24.0	20.0
Ayten O [49]	Turkey	Retrospective	Hospitalized	73	64.4	56.9 ± 13.3	1	0	26	46	45.2	20.5
Puah SH [105]	Singapore	Prospective	Hospitalized	102	73.5	62 (54–68)	1	3	14	84	62.7	37.3
Celik I [57]	Turkey	Retrospective	Hospitalized	160	65.6	53 (24–65)	2	14	37	107	33.1	23.8
Jandaghian S [74]	Iran	Cross-sectional	Hospitalized	4152	56.2	61.10 ± 16.97	10	98	467	3577	33.9	28.9
Ozger HS [98]	Turkey	Prospective	Hospitalized	37	64.9	61 (50–72)	2	3	6	26	54.1	27.0
Kaya T [78]	Turkey	Retrospective	Hospitalized	148	45.3	63.2 ± 16.9	3	7	39	99	45.3	29.7
Ma X [92]	China	Retrospective	Hospitalized	459	55.3	44 (32–54)	0	3	15	441	15.9	9.1
AlBahrani S [34]	Saudi Arabia	Retrospective	Hospitalized	169	60.9	53.1 ± 16.7	0	6	3	160	43.2	12.4
Ro S [109]	Japan	Retrospective	Hospitalized	17	64.7	73.71 ± 21.30	1	1	6	9	47.1	35.3
Deeb A [64]	United Arab Emirates	Retrospective	Hospitalized	1075	90.4	46.0 ± 12.3	2	28	99	946	23.7	31.1
Shah M [118]	Pakistan	Prospective	Hospitalized	250	66.0	54.22 ± 12.56	1	6	56	187	34.0	32.8
Satici MO [114]	Turkey	Retrospective	Hospitalized	272	58.1	64.7 ± 14.7	8	23	78	163	52.7	34.6
Bokhary DH [53]	Saudi Arabia	Retrospective	Hospitalized	656	63.3	50 ± 19.4	2	21	130	503	NA	35.9
Argun Barıs S [45]	Turkey	Retrospective	Hospitalized	213	50.2	50.75 ± 13.61	0	11	6	196	21.6	15.0
Alhamar G [35]	Kuwait	Retrospective	Hospitalized	417	62.8	45.38 ± 17.07	12	29	48	328	29.5	23.3
Alizadehsani R [38]	Iran	Retrospective	Hospitalized	660	56.6	68 ± 14	2	19	100	539	40.2	32.3
Emami A [69]	Iran	Retrospective	Hospitalized	2625	55.5	56.85 ± 18.84	35	210	840	1540	37.7	34.0
Abedtash A [25]	Iran	Retrospective	Hospitalized	180	36.7	67.76 ± 18.72	1	4	70	105	40.0	35.6
Parvin S [101]	Bangladesh	Cross-sectional	Hospitalized	972	64.1	54.47 ± 12.73	15	80	146	731	43.6	42.2
Bakhshwin D [51]	Saudi Arabia	Retrospective	Hospitalized	145	55.2	69.22 ± 8.12	1	5	15	124	41.4	57.9
Zarei J [125]	Iran	Retrospective	Hospitalized	10,657	52.7	55.88 ± 18.46	28	198	1683	8748	5.5	18.3
Elhazmi A [68]	Saudi Arabia	Prospective	ICU patients	1468	74.0	55.9 ± 15.1	37	91	503	837	48.6	54.8
Kuwahara M [88]	Japan	Retrospective	ICU patients	70	71.4	67 (38–84)	4	2	25	39	41.4	59.4
Shesha N [119]	Saudi Arabia	Retrospective	Hospitalized	1583	61.8	50.8 ± 15.8	6	30	172	1375	12.2	19.1
Sener MU [116]	Turkey	Retrospective	Hospitalized	58	70.7	66.5 (57–71)	3	0	32	23	62.1	32.8
Alzahrani MA [43]	Saudi Arabia	Retrospective	Hospitalized	536	53.4	54.3 ± 16.6	5	40	27	464	44.9	44.9
ALGhamdi MA [32]	Saudi Arabia	Retrospective	Hospitalized	248	75.8	49.38 ± 15.46	3	3	42	200	29.8	34.7
Jo S [76]	Korea	Retrospective	Hospitalized	5153	41.5	49.3 (33.2–65.7)	13	109	212	4819	22.2	13.0
Paul G [103]	India	Retrospective	Hospitalized	690	65.4	60.5 (46.7–80.2)	4	2	342	342	38.7	52.2

Abbreviations: ICU, intensive care unit; NA, not available; OR, odds ratio; CI, confidence interval.

## Data Availability

The data that support the findings of this study are included in this article and are available from the corresponding authors upon reasonable requests.

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
