# Peer review of "Comorbid Asthma Increased the Risk for COVID-19 Mortality in Asia: A Meta-Analysis"

_vaccines, 2022, doi:10.3390/vaccines11010089_

Round 1

Reviewer 1 Report

In this meta-analysis the authors have studied the effect of co-morbid asthma on COVID-19 mortality in Asia. The authors claim that this is the largest meta-analysis in Asia on this topic. The study seems to be very well performed, and has included as much as 103 Asian studies in this meta-analysis. However, due to the conflicting data in the literature, this reviewer has some concerns about the interpretation of the results.

MAJOR COMMENTS

1. Many studies, both Asian and non-Asian studies have shown a significantly protective effect of asthma on COVID-19 mortality, and many studies are non-conclusive. Furthermore, the pooled odds ratio found in the present study is relatively low, 1.42 with a lower confidence limit of 1.20. Therefore, it may not be correct to conclude: “Clinicians and nursing staff are supposed to identify and monitor these high-risk patients in a timely manner and provide appropriate clinical treatment for them.” The expression “high-risk patients” seems a little bit too strong based on the results in the present study.

2. As the authors state themselves, asthma is a heterogenous disease, and some phenotypes are related to overweight/obesity (Wenzel, S.E., Asthma phenotypes: the evolution from clinical to molecular approaches. Nat Med, 2012. 18(5): p. 716-25). Furthermore, high BMI has been identified as a risk factor for COVID-19 mortality. This indicates that the authors should also perform subgroup analyses stratified for overweight and/or obesity. Interestingly, in the subgroup analyses stratified according to the proportion of males, the odds ratio increased to 2.08 (lower confidence limit 1.78) in the group with males being less than 50%, and was reduced to 1.24 in the subgroup where males dominated. The lower confidence limit (1.00) in the latter group even indicates borderline significance. This is highly interesting since females usually dominate in obesity-related asthma (Wenzel et al.). Part of the conflicting results on asthma and COVID-19 mortality could be due the different handling of obesity in different studies. This should be discussed.

MINOR COMMENTS

3. Introduction, line 5 from top: “The vaccines and treatment

measures currently developed for COVID-19 were not particularly effective, so it is particularly important to find risk factors leading to disease severity.” This is not entirely correct. Vaccines have shown to be very effective against severe COVID-19 disease and mortality.

4. Statistical analysis, line 3 from bottom: “…by omitting one study once…” should probably be “…by omitting one study at a time…”?

5. Table 1: better with “Dead” than “Death”.

6. Discussion, line 4 from bottom (In conclusion…): “…especially among younger individuals.” Perhaps better with “…at least among non-elderly individuals.”

Author Response

Response to Reviewer 1 Comments

In this meta-analysis the authors have studied the effect of co-morbid asthma on COVID-19 mortality in Asia. The authors claim that this is the largest meta-analysis in Asia on this topic. The study seems to be very well performed, and has included as much as 103 Asian studies in this meta-analysis. However, due to the conflicting data in the literature, this reviewer has some concerns about the interpretation of the results.

MAJOR COMMENTS

Point 1: Many studies, both Asian and non-Asian studies have shown a significantly protective effect of asthma on COVID-19 mortality, and many studies are non-conclusive. Furthermore, the pooled odds ratio found in the present study is relatively low, 1.42 with a lower confidence limit of 1.20. Therefore, it may not be correct to conclude: “Clinicians and nursing staff are supposed to identify and monitor these high-risk patients in a timely manner and provide appropriate clinical treatment for them.” The expression “high-risk patients” seems a little bit too strong based on the results in the present study.

Response 1: Thank you very much for the reviewer’s kind suggestions and comments. All of your valuable comments and suggestions will be helpful to improve the quality of our work. We completely agreed with your comments that based on the relatively low pooled odds ratio found in the present study, the expression of “high-risk patients” seems a little bit too strong. We have removed ”Clinicians and nursing staff are supposed to identify and monitor these high-risk patients in a timely manner and provide appropriate clinical treatment for them” in our revised manuscript according to your valuable comments. Thanks a lot for your valuable comments again.

Point 2: As the authors state themselves, asthma is a heterogenous disease, and some phenotypes are related to overweight/obesity (Wenzel, S.E., Asthma phenotypes: the evolution from clinical to molecular approaches. Nat Med, 2012. 18(5): p. 716-25). Furthermore, high BMI has been identified as a risk factor for COVID-19 mortality. This indicates that the authors should also perform subgroup analyses stratified for overweight and/or obesity. Interestingly, in the subgroup analyses stratified according to the proportion of males, the odds ratio increased to 2.08 (lower confidence limit 1.78) in the group with males being less than 50%, and was reduced to 1.24 in the subgroup where males dominated. The lower confidence limit (1.00) in the latter group even indicates borderline significance. This is highly interesting since females usually dominate in obesity-related asthma (Wenzel et al.). Part of the conflicting results on asthma and COVID-19 mortality could be due the different handling of obesity in different studies. This should be discussed.

Response 2: We greatly appreciated the reviewer’s kind suggestions and comments. Indeed, high BMI has been identified as a risk factor for COVID-19 mortality. We have examined 103 articles included again and found that only 19 studies reported information about BMI or prevalence of obesity. Then we have performed subgroup analyses stratified for obesity according to your valuable comments, which showed that the significant association existed among studies with the prevalence of obesity ≥ 20% (13 studies, pooled OR = 2.27, 95% CI: 1.65-3.12), but did not exist among studies with the prevalence of obesity < 20% (6 studies, pooled OR = 1.21, 95% CI: 0.60-2.46). Part of the conflicting results on asthma and COVID-19 mortality could be due to the different handling of obesity in different studies, we discussed this in our revision according to your valuable comments. Thanks a lot for your valuable comments again.

MINOR COMMENTS

Point 3: Introduction, line 5 from top: ”The vaccines and treatment measures currently developed for COVID-19 were not particularly effective, so it is particularly important to find risk factors leading to disease severity.” This is not entirely correct. Vaccines have shown to be very effective against severe COVID-19 disease and mortality.

Response 3: We are greatly appreciated for the reviewer’s kind suggestions and comments. We completely agreed with your comments that the vaccines and treatment measures currently developed for COVID-19 were not particularly effective is not entirely correct. So we have replaced “The vaccines and treatment measures currently developed for COVID-19 were not particularly effective, so it is particularly important to find risk factors leading to disease severity” with “Vaccines have shown to be very effective against severe COVID-19 disease and mortality, it is also important to understand risk factors (e.g. to decide whom to prioritize for vaccination)” in the introduction according to your kind comments. Thanks a lot again.

Point 4: Statistical analysis, line 3 from bottom: “…by omitting one study once…” should probably be “…by omitting one study at a time…”?

Response 4: We are greatly appreciated for the reviewer’s kind suggestions and comments. We have replaced “…by omitting one study once…”with“…by omitting one study at a time…” in our revised manuscript according to your valuable comments. Thanks a lot for your valuable comments again.

Point 5: Table 1: better with “Dead” than “Death”.

Response 5: Thank you very much for the reviewer’s kind suggestions and comments. We have replaced “Death” by “Dead” in Table 1 according to your valuable comments and suggestions. Many thanks for your kind suggestions.

Point 6: Discussion, line 4 from bottom (In conclusion…): “…especially among younger individuals.” Perhaps better with “…at least among non-elderly individuals.”

Response 6: Thank you very much for the reviewer’s kind suggestions and comments. We have replaced “…especially among younger individuals” with “…at least among non-elderly individuals” in our revised manuscript according to your valuable comments and suggestions. Thanks a lot for your valuable comments again.

Reviewer 2 Report

Estimated Authors,

first of all, thank you for the opportunity to review this very interesting study. in this meta-analysis, Shi et al. have summarised the evidences on the morbidity and mostly mortality associated with COVID-19 in patients affected by Asthma in Asian countries. Albeit very interesting, the present study is affected by significant shortcomings that, from the point of view of the present reviewer, impair the eventual acceptance of this paper, at least in its current stage of development.

More precisely:

a) Asthma is a very heterogenous condition, as this term includes allergic asthma, non-allergic asthma, etc. Moreover, the severity of asthma is in turn quite heterogenous. Eventually, the very same management of Asthma (even when the current guidelines are accurately followed by caregivers) may lead patients to substantial differences in the management according to the economic and healthcare system characteristics of the country. In this study, no information is provided on the background characteristics of the patients included in the study. 

b) Asia is not only the largest and most populated continent of the world, but it is also the more strikingly heterogenous one in economic and demographic terms, as it encompasses very developed areas (e.g. main cities of Mainland China, Japan, South Korea, Singapore, etc.) but also areas economically deprived. As the mortality of Asthma is historically well correlated with the background management of this disorder, being the latter in turn correlated with the economic features of the parent country, not performing appropriate subanalyses by areas characterized by homogenous socioeconomic features may lead to a biased pooled analysis. The sensitivity analysis performed by Study Authors suggests that no single study has reasonably caused a directional change in results, but subgroup analysis would significantly improve the overall quality of this study, as well as for point a) of this review.

c) What about the share of participants that were vaccinated against SARS-CoV-2? Even though Authors state that "vaccines... were not particularly effective..." (against what? mortality? morbidity? hospitalization? please explain or amend according to the available peer-reviewed literature), vaccines have substantially reduced the case fatality ratio and ultimately the COVID-19 related mortality. Without providing this further analysis and without an appropriate discussion of this feature, the present paper if mortally flawed. 

d) how did the Authors exclude studies by duplication of reported case series? this is an appropriate approach, but the criteria applied by Study authors should be explained, at least as an annex, in order to avoid the potential claims for any cherry picking of included studies.

e) please report separately estimates for ICU and non-ICU patients, as it is reasonable that the former may be characterized by more severe features than the latter

Author Response

Response to Reviewer 2 Comments

first of all, thank you for the opportunity to review this very interesting study. in this meta-analysis, Shi et al. have summarised the evidences on the morbidity and mostly mortality associated with COVID-19 in patients affected by Asthma in Asian countries. Albeit very interesting, the present study is affected by significant shortcomings that, from the point of view of the present reviewer, impair the eventual acceptance of this paper, at least in its current stage of development.

More precisely:

  1. a) Asthma is a very heterogenous condition, as this term includes allergic asthma, non-allergic asthma, etc. Moreover, the severity of asthma is in turn quite heterogenous. Eventually, the very same management of Asthma (even when the current guidelines are accurately followed by caregivers) may lead patients to substantial differences in the management according to the economic and healthcare system characteristics of the country. In this study, no information is provided on the background characteristics of the patients included in the study.

Response: Thank you very much for the reviewer’s kind suggestions and comments. All of your valuable comments and suggestions will be helpful to improve the quality of our work. We have rechecked and extracted information about the severity of asthma in terms of background characteristics of the patients included in the study according to your valuable suggestions, while only 3 studies assessed asthma severity according to the guidelines from the Global Initiatives for Asthma (GINA) and provided information about the prevalence of mild and moderate to severe asthma (Ref 40, Ref 87, Ref 112 in our manuscript). Considering the number of eligible studies reporting the severity of asthma is relatively small, we did not assess the impact of asthma severity on COVID-19 mortality presently, which should be a focus in the future study. We addressed this issue in the limitations section in our revision. Thanks a lot for your valuable comments again.

  1. b) Asia is not only the largest and most populated continent of the world, but it is also the more strikingly heterogenous one in economic and demographic terms, as it encompasses very developed areas (e.g. main cities of Mainland China, Japan, South Korea, Singapore, etc.) but also areas economically deprived. As the mortality of Asthma is historically well correlated with the background management of this disorder, being the latter in turn correlated with the economic features of the parent country, not performing appropriate subanalyses by areas characterized by homogenous socioeconomic features may lead to a biased pooled analysis. The sensitivity analysis performed by Study Authors suggests that no single study has reasonably caused a directional change in results, but subgroup analysis would significantly improve the overall quality of this study, as well as for point a) of this review.

Response: Thank you very much for the reviewer’s kind suggestions and comments. According to the suggestions of the reviewer, we have performed further subgroup analysis by country characterized by homogenous socioeconomic features and the results revealed that COVID-19 patients with asthma had a significantly increased risk for mortality compared with patients without asthma among developed countries (pooled OR = 2.06, 95% CI: 1.73-2.47) and developing countries (pooled OR = 1.28, 95% CI: 1.01-1.61), but not among deprived countries (pooled OR = 1.09, 95% CI: 0.67-1.77), which has been added in our revised manuscript. Thanks a lot for your valuable comments again.

  1. c) What about the share of participants that were vaccinated against SARS-CoV-2? Even though Authors state that "vaccines... were not particularly effective..." (against what? mortality? morbidity? hospitalization? please explain or amend according to the available peer-reviewed literature), vaccines have substantially reduced the case fatality ratio and ultimately the COVID-19 related mortality. Without providing this further analysis and without an appropriate discussion of this feature, the present paper if mortally flawed.

Response: We greatly appreciated the reviewer’s valuable suggestions and comments. We have inspected all the literature once more according to the available peer-reviewed literature and found that just 3 studies mentioned vaccine information. Rehman et al.’s study reported that about 13% of infected patients were vaccinated (Ref 71), Shesha et al. and Basaran et al. indicated that participants weren’t vaccinated at the study time (Ref 31 and Ref 54). We have rephrased “vaccines... were not particularly effective...” as “Vaccines have shown to be very effective against severe COVID-19 disease and mortality, it is also important to understand risk factors (e.g. to decide whom to prioritize for vaccination)” in the Introduction section in our revised manuscript according to your valuable comments. Many thanks for your kind suggestions.

d): How did the Authors exclude studies by duplication of reported case series? this is an appropriate approach, but the criteria applied by Study authors should be explained, at least as an annex, in order to avoid the potential claims for any cherry picking of included studies.

Response: Thank you very much for the reviewer’s kind suggestions and comments. Our criteria for excluding duplicate reported case series studies were: If two or more publications are sourced with the same author or the same institute, we then reviewed the time period of participants enrollment among the studies. If the time period of participants enrollment was the same or the study start and end times were crossed among the studies, we regarded these studies had the same participants or overlapping participants, otherwise, we regarded these studies were different. For these studies based on the same data source, we included only the articles with the most complete data. For instance, two studies by Abrishami et al. evaluated the same time period of participant enrollment, so the study with the smaller sample size was excluded (PMID: 32655021; Ref 18). We have amended this criteria in the methods section of revision. Thanks a lot again.

  1. e) Please report separately estimates for ICU and non-ICU patients, as it is reasonable that the former may be characterized by more severe features than the latter.

Response: We greatly appreciated the reviewer’s kind suggestions and comments. We have performed subgroup analysis according to ICU and non-ICU patients, and the results suggested that COVID-19 patients with asthma had a significantly increased risk of mortality among studies for non-ICU patients (pooled OR = 1.45, 95% CI: 1.22-1.72), but not among studies for ICU patients (pooled OR = 1.40, 95% CI: 0.58-3.38). This suggested that disease severity might affect the association between asthma and COVID-19 mortality. Thanks a lot for your valuable comments again.

Reviewer 3 Report

At first impression, this paper is a standard meta-analysis. Generally, the selection process of the papers is well described and the statistical methods seem correct.

But I do have some (mostly minor) concerns that can easily be amended:

Introduction:

“The vaccines and treatment measures currently developed for COVID-19 were not particularly effective, so it is particularly important to find risk factors leading to disease severity.” Maybe it is not wise to state in a vaccines journal, that the vaccines were not effective. And this even without providing a reference for this statement! And besides: If vaccines (or treatment) were effective, would it not also be important to understand risk factors (e.g. to decide whom to prioritize for vaccination)?

“Although there have been several meta-analyses exploring the association between comorbid asthma and the risk for mortality among COVID-19 patients in the full regions [6-10], the findings from previous meta-analyses were still inconclusive, which might suffer limitations from considerable variability in asthma prevalence among different areas.” What do you mean by “regions” and by “areas”? Are these words synonymous? Do you mean continents? Because you carry on arguing that therefore it makes sense to study Asia separately. Which I do not really understand. Yes, asthma is a heterogeneous disease with widely differing prevalence between countries and regions. This could partly be due to different prevalence of (genetic and/or environmental) risk factors for asthma, but also due to medical, cultural and socio-economic differences in diagnostic practice. When in a country only severe asthma cases are diagnosed, a lower prevalence would be reported. But severe cases might be more at risk of death (also during any respiratory infection) than milder cases. But is “Asia” homogeneous as to asthma prevalence and as to underlying medical, cultural and socio-economic characteristics? If you assume that differences between countries (or regions) are the reason for heterogeneity in studies on asthma as risk factor for the outcome of COVID-19, you would better collect studies from all over the world and examine which country- or region-wise differences account best for the heterogeneity. “Asia” in your paper spreads from Turkey and the Arabian Peninsula to Korea. I really don’t see this as a homogenous area or region. You need better arguments for this choice of study focus. In the results you report that you excluded 197 studies because they did not focus on Asia. In my understanding, this is a waste of work (you had already checked the papers!). But of course, the focus on Asia is your decision. Also examining 103 studies is quite a feat!

Results:

Table 1: Why is the first line of the content (Lee SC,…) in bold and underlined? It is not clear to me how the papers were sorted. The reference list uses numbers. It would be helpful to have the same numbers also in the table. Otherwise the reader cannot directly access the full reference.

“Overall combining the data from 103 studies, our meta-analysis indicated that there was a significant association between comorbid asthma and increased risk for mortality of COVID-19 patients (pooled OR = 1.42, 95% CI: 1.20-1.68; I² = 70%, P < 0.01, Fig. 2).” There is a lot of heterogeneity in the studies. This is clear from the test statistic reported here, but also from figure 2. I do miss a clear discussion of this finding. In this sub-chapter you do report stratified effect estimates. You claim that you only found heterogeneity per age-group. Well, I find that there is a remarkable difference between studies that included more women (OR=2.08) than those that included more men (OR=1.24). Now, this stratification would also compare studies with 49% men to studies with 51% men. A better approach would be an analysis stratified by sex in individual studies. I wonder if among the 103 studies non has done this? Differences per gender must be large if even on the group level you find such a strong difference. Did you perform a formal test of interaction? Yes, it seems you did. And this confirmed the influence of male sex.

The differences between age groups are small in comparison: younger than 60: OR=1.44; 60 and older: OR=1.36. Yes, the latter estimate was not significant. But was it significantly different from 1.44?

Figure 2 is ordered by effect estimate. Again, reference numbers would be helpful.

A sensitivity analysis leaving one out at a time, might be a good idea if the number of studies is small. But with 103 studies, the weight of a single study is expected to be not very relevant. I am not sure what figure 3 proves. And a random effect estimate of 103 repeated meta-analyses is also meaningless. I would drop this figure and simply state that the effect estimate was not unduly impacted by any single study.

Discussion:

“Subgroup analyses by male proportion, sample sizes and setting yielded consistent results, but subgroup analysis by age indicated that comorbid asthma was significantly associated with higher risk for COVID-19 mortality in Asia among studies with mean/median age < 60 years old, not among studies with mean/median age ≥ 60 years old.” But your meta-regression showed that only gender, not age, contributed to heterogeneity. No significant effect in older people might simply be an issue of study power. Yes, comorbidities could lead to additional “noise” and measurement error. Especially in studies that did not control for comorbidities. It would be interesting to see how confounder control affected effect estimates.

Author Response

Response to Reviewer 3 Comments

Point 1: Introduction: “The vaccines and treatment measures currently developed for COVID-19 were not particularly effective, so it is particularly important to find risk factors leading to disease severity.” Maybe it is not wise to state in a vaccines journal, that the vaccines were not effective. And this even without providing a reference for this statement! And besides: If vaccines (or treatment) were effective, would it not also be important to understand risk factors (e.g. to decide whom to prioritize for vaccination)?

Response 1: Thank you very much for the reviewer’s kind suggestions and comments. All of your valuable comments and suggestions will be helpful to improve the quality of our work. We have replaced ”The vaccines and treatment measures currently developed for COVID-19 were not particularly effective, so it is particularly important to find risk factors leading to disease severity” with “Vaccines have shown to be very effective against severe COVID-19 disease and mortality, it is also important to understand risk factors (e.g. to decide whom to prioritize for vaccination)” in the introduction of our revision according to your kind comments. Thanks a lot again.

Point 2: Introduction: “Although there have been several meta-analyses exploring the association between comorbid asthma and the risk for mortality among COVID-19 patients in the full regions [6-10], the findings from previous meta-analyses were still inconclusive, which might suffer limitations from considerable variability in asthma prevalence among different areas.” What do you mean by “regions” and by “areas”? Are these words synonymous? Do you mean continents? Because you carry on arguing that therefore it makes sense to study Asia separately. Which I do not really understand. Yes, asthma is a heterogeneous disease with widely differing prevalence between countries and regions. This could partly be due to different prevalence of (genetic and/or environmental) risk factors for asthma, but also due to medical, cultural and socio-economic differences in diagnostic practice. When in a country only severe asthma cases are diagnosed, a lower prevalence would be reported. But severe cases might be more at risk of death (also during any respiratory infection) than milder cases. But is “Asia” homogeneous as to asthma prevalence and as to underlying medical, cultural and socio-economic characteristics? If you assume that differences between countries (or regions) are the reason for heterogeneity in studies on asthma as risk factor for the outcome of COVID-19, you would better collect studies from all over the world and examine which country- or region-wise differences account best for the heterogeneity. “Asia” in your paper spreads from Turkey and the Arabian Peninsula to Korea. I really don’t see this as a homogenous area or region. You need better arguments for this choice of study focus. In the results you report that you excluded 197 studies because they did not focus on Asia. In my understanding, this is a waste of work (you had already checked the papers!). But of course, the focus on Asia is your decision. Also examining 103 studies is quite a feat!

Response 2: Thank you very much for the reviewer’s kind suggestions and comments. In the introduction, “regions” and “areas” mean continents, these words are synonymous. We are very sorry for this confusion. Considering the ambiguity of the previous statement, we have revised the manuscript. According to the suggestions of the reviewer, We have rechecked and extracted information about the severity of asthma and discovered that only 3 studies assessed asthma severity according to the guidelines from the Global Initiatives for Asthma (GINA) and provided information about the prevalence of mild and moderate to severe asthma (Ref 40, Ref 87, Ref 112). Considering the number of eligible studies reporting the severity of asthma is relatively small, we did not assess the impact of asthma severity on COVID-19 mortality presently, which should be a focus in the future study. We addressed this issue in the limitations section of revision. As the reviewer commented that "Asia" is not homogeneous in terms of asthma prevalence and underlying medical, cultural and socio-economic characteristics, we performed subgroup analysis according to the types of country characterized by homogenous socioeconomic features, which suggested that COVID-19 patients with asthma had a significantly increased risk for mortality compared with those without asthma among developed country (pooled OR = 2.06, 95% CI: 1.73-2.47) and developing country (pooled OR = 1.28, 95% CI: 1.01-1.61), but not among deprived country (pooled OR = 1.09, 95% CI: 0.67-1.77), which has been added in our revised manuscript. Thanks a lot for your valuable comments again.

Point 3: Table 1: Why is the first line of the content (Lee SC,…) in bold and underlined? It is not clear to me how the papers were sorted. The reference list uses numbers. It would be helpful to have the same numbers also in the table. Otherwise the reader cannot directly access the full reference.

Response 3: We greatly appreciated the reviewer’s kind suggestions and comments. We are very sorry for this mistake, which has been resolved in our revised manuscript. The papers were sorted according to the order of data extraction. In order to that the reader can directly access the full reference, we have added the reference list’s numbers in the table 1 in our revision according to your kind comments. Thanks a lot for your valuable comments again.

Point 4: Results: “Overall combining the data from 103 studies, our meta-analysis indicated that there was a significant association between comorbid asthma and increased risk for mortality of COVID-19 patients (pooled OR = 1.42, 95% CI: 1.20-1.68; I² = 70%, P < 0.01, Fig. 2).” There is a lot of heterogeneity in the studies. This is clear from the test statistic reported here, but also from figure 2. I do miss a clear discussion of this finding. In this sub-chapter you do report stratified effect estimates. You claim that you only found heterogeneity per age-group. Well, I find that there is a remarkable difference between studies that included more women (OR=2.08) than those that included more men (OR=1.24). Now, this stratification would also compare studies with 49% men to studies with 51% men. A better approach would be an analysis stratified by sex in individual studies. I wonder if among the 103 studies non has done this? Differences per gender must be large if even on the group level you find such a strong difference. Did you perform a formal test of interaction? Yes, it seems you did. And this confirmed the influence of male sex. The differences between age groups are small in comparison: younger than 60: OR=1.44; 60 and older: OR=1.36. Yes, the latter estimate was not significant. But was it significantly different from 1.44?

Response 4: We are greatly appreciated for the reviewer’s kind suggestions and comments. If the heterogeneity disappeared in all subgroup analyses, the stratified variable might be the source of heterogeneity. Meta-regression displayed that male proportion (P = 0.034) and country (P = 0.018) might be the potential sources of heterogeneity and the discussion concerning study heterogeneity has been added in the section of discussion in our revised manuscript. Among the 103 studies, none has done this analysis stratified by sex. Although there is a remarkable difference between studies that included more women (OR = 2.08) than those that included more men (OR = 1.24), the confidence interval for both groups did not include 1. The results for both groups indicated a significant association between comorbid asthma and increased risk for mortality of COVID-19 patients. Although the differences in OR between age groups are small in comparison, 1.36 (point estimate effect) might not be significantly different from 1.44 (point estimate effect) without considering its corresponding confidence interval. Because the confidence interval of point estimate effect among the 60 and older group includes 1, indicating that the significant relationship did not exist. Thanks a lot again.

Point 5: Figure 2 is ordered by effect estimate. Again, reference numbers would be helpful.

Response 5: Thank you very much for the reviewer’s kind suggestions and comments. We have added the reference numbers in the figure 2 in our revision according to your kind comments. Many thanks for your kind suggestions.

Point 6: Results: A sensitivity analysis leaving one out at a time, might be a good idea if the number of studies is small. But with 103 studies, the weight of a single study is expected to be not very relevant. I am not sure what figure 3 proves. And a random effect estimate of 103 repeated meta-analyses is also meaningless. I would drop this figure and simply state that the effect estimate was not unduly impacted by any single study.

Response 6: We are greatly appreciated for the reviewer’s kind suggestions and comments. We completely agreed with your comments that “with 103 studies, the weight of a single study is expected to be not very relevant”. We have dropped figure 3 and simply stated that the effect estimate was not unduly impacted by any single study in our revised manuscript according to your valuable comments. Thanks a lot again.

Point 7: Discussion: “Subgroup analyses by male proportion, sample sizes and setting yielded consistent results, but subgroup analysis by age indicated that comorbid asthma was significantly associated with higher risk for COVID-19 mortality in Asia among studies with mean/median age < 60 years old, not among studies with mean/median age ≥ 60 years old.” But your meta-regression showed that only gender, not age, contributed to heterogeneity. No significant effect in older people might simply be an issue of study power. Yes, comorbidities could lead to additional “noise” and measurement error. Especially in studies that did not control for comorbidities. It would be interesting to see how confounder control affected effect estimates.

Response 7: Thank you very much for the reviewer’s kind suggestions and comments. Our study showed that gender and country contributed to heterogeneity, because the P-value of meta-regression were less than 0.05, but the P-value of age was greater than 0.05. According to the suggestions of the reviewer, we have rechecked and extracted information about adjusted effect estimates and found that 19 studies reported adjusted-OR. Considering that comorbidities could lead to additional ”noise” and measurement error, we subsequently calculated the pooled OR on the basis of adjusted effect estimates. The results indicated that asthma was significantly associated with the increased risk of mortality among Asian COVID-19 patients (19 studies, adjusted-OR = 1.22, 95% CI: 1.05-1.42), which supported the findings which were based on crude effects, which has been added in our revised manuscript. Thanks a lot for your valuable comments again.

Round 2

Reviewer 1 Report

I have no further comments

Reviewer 2 Report

Estimated Authors,

Thank you for having extensively and appropriately addressed all my concerns. 

From my point of view, not only the paper has been radically improved, but a lot of information now you provide in the results sections may be of certain interest for the international readers.

I've only a couple of requirements, and the final acceptance of this study is, from my point of view, only a small step away.

First, please edit figure 2 in order to provide forrest plots for subgroup analysis at least from the point of view of geographical settings; in this regard, please provide across the main text a definition of "developed" vs. "developing" countries; this is explained in the rebuttal letter but it is not specified in the main text. 

Second, the results on ICU vs. non-ICU deaths may be explained by the fact that ICU patients may rely either on a better economic status (theirs or for their countries) than non-ICU patients (whose access to intensive care may be impaired by economic status). Please discuss this topic.

Author Response

Response to Reviewer 2 Comments

From my point of view, not only the paper has been radically improved, but a lot of information now you provide in the results sections may be of certain interest for the international readers.

I've only a couple of requirements, and the final acceptance of this study is, from my point of view, only a small step away.

Point 1: First, please edit figure 2 in order to provide forest plots for subgroup analysis at least from the point of view of geographical settings; in this regard, please provide across the main text a definition of "developed" vs. "developing" countries; this is explained in the rebuttal letter but it is not specified in the main text. 

Response 1: Thank you very much for the reviewer’s kind suggestions and comments. All of your valuable comments and suggestions will be helpful to improve the quality of our work. We have re-edited figure 2 in our revised manuscript according to your valuable comments. Additionally, we characterize the country’s levels of social and economic development based on the 4-tier Human Development Index (HDI) from the United Nation's 2022 Human Development Report (https://hdr.undp.org/content/human-development-report-2021-22). Countries were also categorized into very high HDI country (Korea, Singapore, Israel, Japan, Turkey, Kuwait, Saudi Arabia and United Arab Emirates), high HDI country (China, Iran and Indonesia), medium HDI country (India, Philippines and Bangladesh) and low HDI country (Pakistan), which has been added in the section of results in our revised manuscript according to your valuable comments. Further subgroup analysis by country characterized by homogenous socioeconomic features revealed that COVID-19 patients with asthma had a significantly increased risk for mortality compared with patients without asthma among very high HDI country (pooled OR = 1.55, 95% CI: 1.29-1.87, Fig. 2), but not among high HDI country (pooled OR = 1.38, 95% CI: 0.91-2.11, Fig. 2), medium HDI country (pooled OR = 0.88, 95% CI: 0.50-1.56, Fig. 2) and low HDI country (pooled OR = 1.60, 95% CI: 0.77-3.33, Fig. 2). Thanks a lot for your valuable comments again.

Point 2: Second, the results on ICU vs. non-ICU deaths may be explained by the fact that ICU patients may rely either on a better economic status (theirs or for their countries) than non-ICU patients (whose access to intensive care may be impaired by economic status). Please discuss this topic.

Response 2: We greatly appreciated the reviewer’s kind suggestions and comments. We have discussed this topic in our revision according to your valuable comments. Thanks a lot for your valuable comments again.